# Influenza Vaccination of Nursing Students: A Cross-Sectional Study of Uptake, Knowledge, Attitudes, and Practices in Greece

**DOI:** 10.3390/diseases12080183

**Published:** 2024-08-14

**Authors:** Anastasia Statiri, Theodoula Adamakidou, Nikoletta Margari, Ourania Govina, Chrysoula Tsiou, Konstantinos Giakoumidakis, Eleni Dokoutsidou

**Affiliations:** 1Department of Nursing, University of West Attica, 12243 Athens, Greece; thadam@uniwa.gr (T.A.); nmargari@uniwa.gr (N.M.); ugovina@uniwa.gr (O.G.); ctsiou@uniwa.gr (C.T.); edokout@uniwa.gr (E.D.); 2Department of Nursing, Hellenic Mediterranean University, 71410 Heraklion, Greece; kongiakoumidakis@hmu.gr

**Keywords:** influenza, nursing students, vaccine uptake, knowledge, attitudes, practices, Greece

## Abstract

Influenza immunization includes a yearly repeated vaccine offered to every healthcare worker, including nursing students, with a high risk of contracting this viral disease. This study aimed to investigate the vaccination coverage, knowledge, attitudes, and practices of nursing students against influenza in Greece. A cross-sectional study was conducted in the Attica region between September 2022 and July 2023, with the use of an anonymous reference questionnaire. Data from 1261 nursing students were recorded (response rate: 68.6%). The study found that 23% of the sample were vaccinated against influenza for the flu season 2022–2023, and 42% were vaccinated for the previous flu season. Knowledge scores regarding influenza ranged from 0% to 100%, with a mean value of 55 (SD = 18.8%). A higher level of knowledge about influenza was associated with more appropriate attitudes and practices toward the disease (*p* < 0.001). Notably, participants in their second, third, or fourth year of study and beyond exhibited more suitable attitudes and practices towards the flu compared to those in their first year of study (*p* < 0.05). The emergence of low vaccination coverage identifies the need for departments of nursing studies to proceed with the design of educational and intervention programs on infection control.

## 1. Introduction

Influenza pandemics have occurred approximately every decade since 1889, with the most recent in history being in 2009. The enormous loss of life and the accompanying economic damage have given a strong impetus to science to seek preventive measures to prevent future deadly pandemics [1].

Immunization against the influenza virus lasts approximately six months, and therefore, vaccination should be repeated annually, even if the influenza strains that were prevalent in the previous year are the same as in the current year of vaccination. The composition of the vaccine changes every year following the needs of the season, and the composition of the vaccine is recommended by the World Health Organization (WHO) [2].

Vaccination coverage is an indicator of a country’s health and reflects the overall level of public health services provided. The international health organizations’ target for the vaccination of the population against the influenza virus is to exceed 75% of the total [3]. When it comes to healthcare professionals, including nursing students, influenza vaccination is considered mandatory [4]. Nursing students are a special subgroup of healthcare workers (HCWs) who share their study time between clinical and academic environments. Students are at high risk of receiving influenza due to their lack of experience and high knowledge of infection control [5]. Nursing students come into contact with patients during their clinical rotations, as well as with family members and other people in the community where they live. These contacts highlight the importance of annual influenza vaccination for nursing students in breaking the chain of viral transmission.

By the end of 2019, a Polish study took place at the Medical University of Łódź. A total of 1137 healthcare students participated, and 446 of them were nursing students. Only 30.7% of the nursing students (n = 137) were vaccinated for the flu season 2019/2020, and only 1.3% (n = 6) were regularly vaccinated annually [6]. Another small study from Poland revealed that of 470 healthcare students, only 15 were vaccinated against influenza (5.84%), although the 2012/2013 flu season was characterized as endemic. The sample contained only 15 nursing students, and only 1 was vaccinated (6%) [5].

Regarding Sweden, a survey conducted during the 2009/2010 flu season found that 139 out of 164 nursing students (84.8%) were vaccinated against influenza. A significantly higher (*p* < 0.01) proportion of the medical students who participated in the study had been vaccinated compared to the nursing students (93.2% vs. 84.8%). However, nursing students reported protecting their patients from the flu as a main reason for vaccination to a significantly greater extent than medical students, 69% vs. 59%, respectively (*p* < 0.05) [7].

Regarding data from Spain, a cross-sectional study was performed during the 2014–2015 academic year, and out of 227 nursing students who participated in the study, only 5.3% (n = 12) of them were vaccinated against the flu. The main reasons for not receiving the vaccine were the difficulty finding someone to vaccinate the students (59.2%) and the belief that they were not at risk of the disease or its complications (40.8%) [8]. Another cross-sectional study took place with 1122 Spanish nursing students from two different Spanish universities during the academic year 2018–2019. The flu vaccine was accepted by the vast majority of the sample (83.4%, n = 936). Nursing students had good knowledge of vaccines, especially those in the upper academic years, and positive attitudes toward flu vaccination [9]. The same year, another study from Madrid among 624 nursing students showed that only 32.5% (n = 203) of the participants were vaccinated against the seasonal flu [10]. A group of students from the Faculty of Nursing at the University of Madrid participated in a cross-sectional study. The researchers had a hypothesis that vaccination rates would significantly increase due to the COVID-19 pandemic. The study included students who were enrolled in the academic year 2019/2020 before the pandemic began and those enrolled in the year 2021/2022, 18 months after the pandemic was declared. A total of 1336 nursing students took part in the research, 624 during the 2019/2020 campaign and 712 during the 2021/2022 campaign. Influenza vaccination coverage increased significantly (*p* = 0.008) in nursing students between the two campaign seasons. From 203 nursing students (32.5%) during the 2019/2020 campaign to 278 vaccinated nursing students (39%) during the 2021/2022 campaign [11].

The University of Palermo conducted a cross-sectional study with the participation of 403 nursing students. Italians declare that the “ideal” vaccination coverage objective is 95%, while the “minimum” target is 75%. Although 62.53% (n = 252) of the participants considered themselves to have a high risk of contracting infectious diseases such as influenza, only 21.09% (n = 85) had been vaccinated against seasonal flu during 2018/2019. A study conducted in Milan during the same flu season (2018/2019) among 101 nursing students also revealed a low vaccination coverage of 30.7% (n = 31) [12]. The low vaccination coverage for this flu season (2018/2019) is confirmed by a similar study from Rome. Thirty-five out of sixty-one nursing students (57.4%) were immunized against influenza. However, 83.6% of the respondents (n = 51) intended to receive the flu vaccine in the following year [13]. A previous study from the 2017/2018 flu season examined 1035 nursing students. The vast majority of the sample (84.25%, n = 872) had never received a flu vaccine. A total of 4.93% (n = 51) of the participants were once vaccinated against influenza, 4.25% (n = 44) were twice vaccinated, 1.35% (n = 14) were three times vaccinated, and only 5.22% (n = 54) had been vaccinated against influenza more than four times [14].

A study from France revealed that 36 nursing students (16.9%) out of 213 who participated had been vaccinated for the flu season 2011/2012 [15]. Another study from Paris revealed that 31 out of 147 nursing students (21.1%) who participated in the research were vaccinated against influenza for the 2008/2009 flu season [16].

Studies from the United Kingdom revealed a rather low immunization against influenza. A cross-sectional survey among future nurses for the flu season 2008–2009 at the University of Birmingham revealed that only 10 out of 79 nursing students (12.7%) who participated had been vaccinated against the seasonal flu [17]. Another study from the University of Nottingham examined 430 nursing students, 27.6% (n = 118) of whom reported ever having been vaccinated against influenza, 12.2% (n = 57) reported being vaccinated regularly, and only 19.8% (n = 83) indicated that they intended to have the vaccine in advance of winter 2010/2011 [18].

In Greece, the influenza virus is a disease with a mandatory declaration to the National Public Health Organization (NPHO) and is monitored weekly from October to March by six systems nationwide [19]. Influenza vaccine is offered free of cost in Greece, and as far as HCWs are concerned, vaccination is strongly recommended and not mandatory. Although the implementation of mandatory vaccination policies creates ethical concerns and is not broadly accepted by the HCWs, no other measure seems enough to increase compliance [4]. Several studies have been conducted to investigate immunization coverage of nursing students against influenza since they are considered to be at high risk of infection due to traineeships and/or internships in hospital units [7,8,14,15,16,17,18]. However, Greece lacks previously published data regarding flu vaccination coverage for nursing students. The only study from Greece revealed a low vaccination coverage against influenza among nursing students. Out of 136 participants in the region of Athens, only 23 of them (16.9%) had received at least one shot of the vaccine in the past. However, the vast majority of the participants (84.7%) would like a vaccinated healthcare worker to take care of their family members [20]. 

The above findings reveal a wide range of influenza vaccination rates among nursing students across different countries and flu seasons. In some cases, vaccination rates are notably low, such as in Poland (30.7% in 2019/2020) [5] and France (16.9% in 2011/2012) [15], while other countries like Sweden report higher rates (84.8% in 2009/2010) [7]. There are also noticeable differences in vaccination rates not only between countries but also within countries over time. For example, in Spain, the acceptance of the flu vaccine increased from 5.3% in 2014/2015 [8] to 83.4% in 2018/2019 [9], yet another study from the same year showed only 32.5% coverage among Madrid nursing students [10]. In Italy, rates varied significantly between cities and years, such as 21.09% in Palermo (2018/2019) [12] and 57.4% in Rome [13]. Despite the known risks associated with influenza, especially for HCWs, the data show that many nursing students are not regularly vaccinated [8,12,20]. For instance, only 1.3% of nursing students in Poland were regularly vaccinated annually [5], and similar trends of low regular vaccination are observed elsewhere [8,9,15,20]. The reasons for low vaccination coverage are varied and include logistical barriers (e.g., difficulty finding someone to administer the vaccine in Spain [8]), perceptions of risk [9], and beliefs about the vaccine’s importance [7,8]. In Greece, while there is no mandatory vaccination policy, the availability of free vaccines has not translated into high uptake, indicating potential cultural or educational barriers [20]. The findings reflect an ethical dimension, particularly concerning the students’ intentions and preferences regarding patient safety. For instance, in Sweden, a significant portion of nursing students vaccinated themselves to protect patients, highlighting a professional ethic even among students [7]. The data also suggest that significant health events, such as the COVID-19 pandemic, may influence vaccination behavior. For example, the vaccination coverage among nursing students in Spain increased from 32.5% in 2019/2020 to 39% in 2021/2022 [11], possibly due to heightened awareness and concern about infectious diseases.

These findings suggest that while there is awareness of the importance of flu vaccination among nursing students, actual vaccination rates remain suboptimal in many regions. This discrepancy could be due to a combination of logistical, cultural, and educational factors. To address the critical need for current data on influenza immunization among nursing students in Greece, we investigated the uptake of influenza vaccination in this group and examined their knowledge, attitudes, and practices towards influenza. Further objectives of the present study were as follows: (1) To investigate whether socio-demographic characteristics of nursing students and the year of study affect their vaccination coverage regarding influenza. (2) To study any possible relationship between the level of knowledge, attitudes, and practices of nursing students and their flu vaccine uptake. (3) To determine whether influenza level of knowledge is associated with nursing students’ attitudes and practices.

## 2. Materials and Methods

### 2.1. Study Design and Participants

A cross-sectional study was conducted with the participation of all nursing students from the two largest University Institutions in Greece, the University of West Attica (UNIWA) and the National and Kapodistrian University of Athens (UOA), from September 2022 to July 2023. Entry criteria in the present study were attending a Department of Nursing studies and informed consent of the participants. Participants could have any gender, racial or ethnic background, or health condition, and they had to identify themselves as adults aged 17 or older. In addition, a prerequisite for admission to the study was a good knowledge of the Greek language. Exclusion criteria were under 17 years old, inadequate knowledge of the Greek language, and students from departments other than nursing. In total, 1838 nursing students who met the criteria for admission to the study were approached, and 1261 people participated. The response rate reached 68.6%

Students were approached by the research team, who, in the presence of the professor in charge during academic classes, informed the students about the purposes of the study, the voluntary nature of participation, the possibility of withdrawal at any stage of the research, the commitment to respect the confidentiality of students’ data and the mandatory nature of consent before starting to complete the questionnaire. Students completed the self-report questionnaire after their informed consent.

### 2.2. Questionnaire

The survey tool was a structured questionnaire created for the purpose of this study and based on data from the international scientific literature. More specifically, a literature review on students’ knowledge and attitudes regarding influenza vaccination, as well as other relevant assessment tools, was carried out [21,22]. The experience of researchers/professors in communicable diseases, infection prevention, public health, and community nursing and a statistical scientist, as well as discussions with students on the subject, contributed significantly. The resulting questionnaire had 68 questions, and answers were on a 5-point Likert scale. An expert panel consisting of four professors from the nursing field (T.A., O.G., N.Μ., and E.D.) and four nursing specialists in infection prevention and management were informed about the purpose of the tool, and after studying the questionnaire, they were asked to give their comments on the clarity and appropriateness of the questions. The experts assessed each item on the questionnaire as “essential”, “useful but inadequate”, or “unnecessary”. All expert comments were taken into account, and eventually, with the assistance of a statistical scientist, 25 items were eliminated from the questionnaire. The drafting of the questionnaire was followed by a pilot test.

The final questionnaire included socio-demographic characteristics, general attitudes toward vaccination, and, in particular, knowledge, attitudes, practices, and vaccination coverage regarding the influenza virus. 

More specifically, the demographic characteristics of the participants included 13 questions on gender, age, religion, nationality, marital and employment status, insurance capacity, place of permanent residence, university, and year of study, as well as smoking and alcohol consumption habits. 

Students’ general attitudes towards vaccinations included a total of 9 questions. More specifically, 5 questions concerned coverage against key communicable diseases such as pertussis and measles. These questions were structured to elicit a ternary response set, comprising “yes”, “no”, and “don’t know/don’t answer” as possible answers, thereby acknowledging affirmative, negative, and uncertain or unknown positions. A 5-point Likert scale was used to receive answers to 3 questions based on the reliability of the information provided by the Ministry of Health on vaccinations and the reasons for acceptance and non-acceptance of vaccination. The question regarding the last time of vaccination against diphtheria and tetanus received answers in the form of multiple choice.

The questionnaire was followed by 9 questions about the influenza virus, and more specifically, 4 questions received answers in the form of multiple choice (i.e., “What are common complications of the flu virus?”), and 5 questions were structured to elicit a ternary response set, comprising “yes”, “no”, and “don’t know/don’t answer” as possible answers (i.e., “Could influenza have serious and irreversible effects on the health of the general population?”).

Four questions of the “yes”, “no”, and “don’t know/don’t answer” forms were used to retrieve information on students’ attitudes regarding the flu. Participants’ practices toward the disease were examined with 8 questions. The first 5 of them received answers to the “yes”, “no”, and “don’t know/don’t answer” forms. An example could be the question, “Do you consider nursing patients with influenza a problem?”. In this particular question, if the answer given was positive, the participant should answer an extra question explaining the reason for this practice. The last two questions of this part of the tool were multiple choice.

The last part of the questionnaire consisted of 4 questions regarding the vaccination coverage of the students, who had to answer whether they were vaccinated for the present and previous years of influenza and explain the reasons for their answers on a Likert scale. The time to complete the questionnaire was about 15 to 20 min.

### 2.3. Pilot Study

The drafting of the questionnaire was followed by a reliability and validity check. Initially, a pilot test was carried out on a sample of 30 nursing students who were not included in the final sample of the study. The reliability of the research tool was evaluated using Cronbach’s alpha coefficient and test-retest analysis, and the result was 0.723 for the whole questionnaire. Internal reliability for the three parts of the tool (knowledge, attitudes, and practices) was above 0.70, and the test-retest reliability for these parts was significant at *p* < 0.05, showing good stability of the instrument.

### 2.4. Statistical Analysis

The statistical analysis was performed using the SPSS 26.0 software package (SPSS Inc., Chicago, IL, USA) [23]. In cases where participants did not fully answer all items of the research tool, missing values were recorded. The presentation of variables included counts, proportions, mean values (standard deviation), and median (interquartile range) for quantitative variables, as well as absolute and relative frequencies for categorical variables. To identify independent factors associated with influenza vaccination, logistic regression analysis was conducted using a stepwise method (*p* for entry 0.05, *p* for removal 0.10). Adjusted odds ratios (OR) with 95% confidence intervals (95% CI) were calculated based on the logistic regression results. The internal consistency reliability was assessed using Cronbach’s coefficient. All *p* values reported were two-tailed, and statistical significance was defined as *p* < 0.05.

Subsequently, based on the correct responses regarding attitudes and practices towards influenza, percentage scores for recommended practices were calculated. For questions with a single correct answer, this received a score of 1; for multiple-choice questions, each correct answer corresponded to one point, and for continuous variables, higher scores indicated better attitudes and practices. Scores were transformed into percentages so that higher scores reflected more recommended attitudes towards influenza.

### 2.5. Ethics

In accordance with the principles outlined in the Declaration of Helsinki, the study was carried out and received approval from the Research Ethics Committee at the University of West Attica (protocol code 17432/23-02-2022). Prior to their participation, all subjects provided informed consent. The participants were informed in the introductory note of the questionnaire about their anonymity, and only after their written informed consent was the completion of the self-report questionnaire possible.

## 3. Results

Data from 1261 nursing students were recorded. Most students were women (78.8%), and their mean age ± SD was 21.9 ± 5.6 years. Moreover, most of the participants were Greeks (93.2%), Christians (88.8%), singles (87.8%), and unemployed (61%). Furthermore, most students (28.3%) were in the third year of their studies, 24.3% smoked, and 42.1% consumed alcohol. Participants’ characteristics are presented in Table 1. More than half of the sample (63.3%) trusted or trusted completely the reliability of the information provided by the Ministry of Health about vaccinations. 

During the flu season 2022–2023, the percentage of vaccination was 23% (n = 290), while in previous years, it was 42% (n = 529). The most significant reason for influenza vaccination was personal protection (mean 1.2 ± 0.6), followed by the protection of the family environment (mean 1.9 ± 0.5), and lastly, the protection of patients (mean 2.7 ± 0.6). Conversely, the most significant reason for not getting vaccinated did not belong to a high-risk group (mean 1.6 ± 1.0), followed by the belief that they would not get sick (mean 2.5 ± 0.9), adverse events (mean 2.8 ± 1.0), and lastly, vaccine efficacy (mean 3.1 ± 0.9).

Information on participants’ knowledge of influenza is presented in Table 2. The percentage of correct answers ranged from 4.2% to 95.7%. More specifically, 9.1% of the sample correctly answered that postpartum women were considered to be at high risk of contracting the flu, and 4.2% did not know if any of the mentioned groups of people were at high risk of getting the disease. On the other hand, 95.7% of the sample correctly answered that the influenza virus could be transmitted by droplets, and 77.5% said that the flu could be transmitted through contact with a contaminated surface. All correct answers were summed up, and their sum was converted into a percentage. Thus, students’ knowledge scores could range from 0% to 100%, with greater values indicating greater knowledge. Knowledge scores ranged from 0% to 100%, with a mean value of 55% (SD = 18.8%). Cronbach’s alpha reliability index was 0.74, indicating acceptable reliability of the score.

Students’ attitudes and practices on influenza are presented in Table 3. Almost all students (95.6%) used clean gloves in manipulations with an increased possibility of contact with the patient’s bodily fluids, and 94.9% wore a surgical or other protective mask to enter a flu patient’s ward. Also, the vast majority of the sample (90%) had not participated in an influenza-related education program in the past 12 months, and 82% of the nursing students did not consider it a problem to hospitalize patients with influenza. 

After multiple logistic regressions were conducted, it was found that students who were part-time or full-time employed had a 35% and 56% lower probability, respectively, of being vaccinated against influenza compared to unemployed students. Also, higher trust in the reliability of the information provided by the Ministry of Health about vaccinations and having participated in an influenza-related education program in the past 12 months were statistically significantly associated with a greater probability of being vaccinated against influenza (*p* = 0.001). Multiple logistic regression results are presented in Table 4. 

Based on the correct responses regarding attitudes and practices towards influenza, percentage scores for recommended practices were calculated. In this sample, the minimum score was 0, and the maximum was 100. For recommended influenza practices, the mean was 77.6 ± 18.6.

The score of appropriate attitudes and practices about influenza was significantly related to the protection of the nursing students’ family environment (*p* = 0.033), the mandatory vaccination in their workplace (*p* = 0.001), and the outbreak of an epidemic (*p* = 0.001). Those who attached more importance (lower score) to the protection of their family environment also had more appropriate attitudes and practices towards the flu (*p* = 0.033). Likewise, for those who consider vaccination important during an outbreak/pandemic (*p* = 0.001). On the contrary, those who were vaccinated more due to obligation from their work had less appropriate attitudes and practices towards influenza (*p* = 0.001). Influenza knowledge score was found to be significantly associated with mandatory workplace vaccination. Those vaccinated more due to obligation from work had less knowledge about influenza (*p* = 0.040). Finally, reasons for not accepting influenza vaccination were not found to be related to participants’ knowledge and attitudes. 

Another important finding of the present research was that the knowledge score about influenza and the year of study of the participants were found to be independently related to the nursing students’ scores of attitudes and practices towards influenza. Specifically, those participants studying in their second, third, or fourth year of studies and above had more appropriate attitudes and practices towards the flu compared to those who were in their first year of study (*p* < 0.05). The above findings are presented in Table 5. Furthermore, more knowledge about influenza was associated with more appropriate attitudes and practices towards the disease (*p* < 0.001). 

## 4. Discussion

The present study aimed to investigate the influenza vaccine uptake of nursing students, as well as the level of knowledge, the intentions to be vaccinated, and the practices against the virus. Numerous research studies have emphasized the crucial role of nursing students’ knowledge, attitudes, and practices towards vaccines, not only in their own immunization but also in providing effective care to their patients [5,6,11].

Our study found that nursing students with jobs were less likely to get vaccinated than unemployed students. Furthermore, believing health ministry information and attending flu education programs increased vaccination rates. Students who got vaccinated to protect their families had better attitudes toward flu prevention than those vaccinated for work. Additionally, students in later years of study had better flu prevention knowledge and practices compared to first-year students. Overall knowledge about influenza was also linked to better flu prevention behaviors. Lastly, vaccination rates among students dropped significantly during the 2022–2023 flu season.

After multiple logistic regressions were conducted, students who had participated in an influenza-related education program in the past 12 months had a greater probability of being vaccinated against influenza. The early information of the students about the disease and the value of vaccination against it seems to have a positive effect on their acceptance or non-skipping of vaccination. This finding is also confirmed by a recent study by Queen’s University Belfast, which took place in the 2018/2019 flu season [24]. The university applied an intervention program to improve students’ knowledge and perceptions, and the post-intervention vaccination uptake increased from 36.7% to 47.8% in the sample. The above findings underscore the crucial role of education in promoting vaccine uptake. By providing students with accurate and up-to-date information about influenza, its transmission, and the benefits of vaccination, educational programs can effectively address misconceptions and encourage vaccine acceptance. The comparison with the Queen’s University Belfast study further strengthens the argument [24]. While the intervention program in that study led to a modest increase in vaccination rates, it still supports the overall trend that educational efforts can positively influence vaccine uptake.

The present study found that nursing students who were vaccinated more due to obligation from their work had less appropriate attitudes and practices towards influenza (*p* = 0.001). On the same page, working students had a lower chance of being vaccinated compared to unemployed students in our study sample (*p* < 0.05), probably due to lack of time to receive the vaccine. During the 2009/2010 endemic flu season, 56.1% (n = 192) of nursing students from the United States of America were vaccinated [25]. The main obstacle referred by the participants to receiving vaccinations was the locations where vaccinations took place [25]. In Greece, there have been no interventions or on-site vaccination programs in the departments of nursing studies. However, in Australia, specifically at the University of Notre Dame, the acceptance of influenza vaccination among nursing students improved significantly after a low-cost promotional campaign by the students themselves [26]. The contrast between the US and Australian experiences is instructive. The US study emphasizes the importance of convenient vaccination locations [25], while the Australian example demonstrates the effectiveness of student-led initiatives in promoting vaccine uptake [26].

Possibly, similar programs will mobilize nursing students from other countries in the direction of the annual flu vaccination. Nursing schools should consider the vaccination of their students within the academic environment, not only for the sake of saving their working students’ time but also since, in the present study, higher confidence in the reliability of the information provided by official bodies such as the Ministry of Health about vaccinations was positively and statistically significantly associated with a greater likelihood of being vaccinated against influenza. The positive correlation between trust in health authorities and vaccination rates reinforces the need for clear, consistent, and credible communication about influenza and its prevention. Building public trust is essential for successful vaccination campaigns.

Greek nursing students reported protecting their family environment as a main reason for vaccination, and those students also had more appropriate attitudes and practices towards the flu. Greek nursing students prioritize protecting their family environment, reflecting not only the importance of family in Greek culture but also the cultural influence on vaccination motivations in general. Swedish nursing students reported that patient protection from the flu is the main reason for vaccine uptake [7]. For Polish nursing students, the two most common reasons for vaccination were individual protection and decisions made by their parents [6,27]. The findings underscore the importance of considering cultural and societal factors when designing influenza vaccination campaigns. The emphasis on family protection among Greek nursing students suggests that messaging focused on familial well-being might be particularly effective in this population. Conversely, highlighting the role of HCWs in protecting patients might resonate more with Swedish students. The diverse reasons for vaccination among different populations emphasize the need for tailored interventions. Addressing specific misconceptions and concerns can improve vaccine acceptance rates.

Findings from an Italian study suggest that 35.54% of the participating nursing students did not consider themselves at risk of contracting the flu, 16.36% forgot to be vaccinated, 10.02% believed that the vaccine is not effective, 9.13% and 2.84% did not consider themselves to be a source of infection for their families, or for their patients, respectively [28]. These findings highlight the importance of educational campaigns that correct misinformation about influenza and its prevention.

Our research showed that a higher level of knowledge about influenza was associated with more appropriate attitudes and practices towards the disease. This association is reasonable, given that a better understanding of influenza’s risks and benefits of vaccination logically motivates nursing students to be vaccinated. An Israeli study confirmed the above findings [29]. The Bombay Medical College attributed its students’ low vaccination coverage to Indian nursing students’ low level of knowledge about vaccine-preventable diseases such as influenza [30] and also confirmed the association between knowledge and influenza vaccine uptake. Moreover, the Polish study of Kałucka et al. (2020) declared a poor understanding of the difference between influenza and the common cold and a low prevalence of flu vaccination among all healthcare students (nursing, midwifery, pharmacy, and public health) [6]. The consistent findings across different populations underscore the critical role of knowledge in shaping influenza vaccination behaviors. A better understanding of influenza’s risks, the benefits of vaccination, and the distinction between influenza and the common cold are associated with increased vaccine uptake. Consequently, academic and governmental bodies must emphasize educational interventions to improve knowledge about influenza and its prevention.

In addition, the rates of influenza vaccination showed an increase with the progress of the study year of the Greek students. This finding is consistent with the previous one regarding the level of knowledge about influenza and the consequent acceptance of vaccination. Studies in Nursing Science clearly increase students’ knowledge of communicable diseases, such as influenza. More specifically, nursing students in the second, third, or fourth year of study and above had more appropriate attitudes and practices towards influenza compared to those in the first year. The above data were confirmed by a recently published study from Spain [9] and were also applicable to Javier et al.’s (2021) study, confirming that increased knowledge in more senior academic years reflects the above-mentioned relationship between knowledge and attitude [10]. A senior nursing student was the variable that had the greatest positive influence on influenza coverage in Ajejas Bazán et al.’s study (2022) [11]. The aforementioned data consistently indicate that the nursing curriculum effectively contributes to developing informed healthcare professionals who understand the importance of influenza prevention. Moreover, the findings imply that the cumulative effect of nursing education positively influences students’ decision-making regarding vaccination, shaping not only the students’ knowledge but also their practices. The progression through the curriculum provides opportunities to deepen knowledge, enhance critical thinking, and develop a professional identity that prioritizes patient care and public health.

As far as vaccine uptake is concerned, 23% (N = 290) of the nursing students who participated in our research had been vaccinated for the 2022–2023 flu season, and 42% (N = 529) had been vaccinated in previous years. Compared to previously published data from Greece, the flu vaccination coverage of nursing students seems to have remained at consistently low levels since 2014, when we have the latest published data for nursing students, where only 16.9% of students declared vaccinated against influenza [20]. Although influenza vaccination is free of cost in Greece, the optional nature of vaccination for nursing students and other HCWs, except those working in the ICU, infectious units, dialysis, and oncology departments, is one prominent reason for the low vaccine uptake [4]. Studies from Sweden and Spain showed higher vaccination coverage rates of nursing students against influenza, 84.8% [7] and 83.4% [9], respectively. Both countries have an optional influenza vaccination program for HCWs. However, Swedish nursing students consider the protection of their patients as their main reason for vaccination [7], and Spanish nursing students have high levels of knowledge about vaccines and positive attitudes towards influenza vaccination in particular [9]. Similar percentages to the Greek Universities were presented by the Polish University in Łódź, where only 30.7% of students were vaccinated in the 2019/2020 flu season [6]. Poland, like Sweden and Spain, strongly suggests influenza vaccination for all HCWs, including nursing students. However, HCWs in Poland have a slightly higher vaccination rate than the general population. Estimates suggest that nurses have a vaccination rate ranging from 5% to 10% [27].

An additional worrisome aspect of the consistently low rates of influenza vaccination among nursing students in Greece and other countries is the potential impact on patient care. The optional nature of the vaccination policy for HCWs seems to be a significant factor contributing to this problem. However, the imposition of mandatory vaccination regimes can be seen as a limitation on individual liberty, as it restricts personal autonomy over one’s own body. The disparity in vaccination rates between Greece and countries like Sweden and Spain implies that cultural influences, such as the emphasis on patient safety, as well as educational campaigns aimed at enhancing awareness and attitudes towards vaccination, may also play a role in determining vaccine acceptance. Thus, while Poland has guidelines recommending influenza vaccination for all HCWs, the low overall vaccination rate in the general population appears to be influencing the vaccination rates among healthcare professionals [6]. This suggests that increasing vaccination coverage in the general population could indirectly impact vaccination rates among HCWs.

According to French law, since 2008, nurses have been allowed to vaccinate people (except children and pregnant women) against influenza without a medical prescription if individuals have been vaccinated at least once by a doctor [15]. A study conducted by Desbouys et al. (2016) revealed that implementing a law could potentially improve access to influenza vaccination for high-risk patients (77.5%), increase vaccination coverage (53%), save money for the health insurance system (52.1%), raise awareness among nurses about their responsibilities in preventing influenza (51.2%), and promote collaboration and task delegation among HCWs (46.9%) [15]. Additionally, since 2021, influenza vaccination has been compulsory only for health professionals working in hospitals, elderly care units, childcare, and maternity facilities, as well as for nursing students undertaking an internship in these facilities. Although the French government has taken drastic and positive steps in favor of vaccination, the results from the French literature are disappointing. More specifically, 21.1% of nursing students were vaccinated for the 2008/2009 flu season [16], and 16.9% of students vaccinated for the 2011/2012 flu season [15]. The French government’s decision to expand the role of nurses in influenza vaccination is a positive step towards improving access to care and potentially increasing vaccination coverage. The anticipated benefits identified in the Desbouys et al. (2016) study are promising [15]. The example of France is a best practice and can be adopted by other countries, including Greece, with the aim of increasing the vaccination coverage of the population. However, the persistently low vaccination rates among nursing students, despite these policy changes, suggest that additional strategies are needed to promote vaccine uptake. The limited scope of the mandatory vaccination policy may also contribute to the low vaccination rates among nursing and other healthcare professional students. While mandatory vaccination for specific healthcare settings is important, a broader and multifaceted consideration regarding students’ vaccination rates could potentially have a more significant impact on overall vaccination coverage.

In the United Kingdom (UK), influenza vaccination is strongly recommended in the same facilities as in France and simply suggested for the rest of the HCWs and nursing students. Only 12.7% of the nursing students of the University of Birmingham had received influenza vaccination in the 2008/2009 influenza season [17]. Another study from the University of Nottingham reported that 27.6% of nursing students had never received the flu vaccine [18]. The less stringent vaccination recommendations in the UK compared to France may partially explain the lower influenza vaccination rates among nursing students in the UK. The low vaccination rates among nursing students in the UK are concerning, as they highlight a gap in protecting both students and patients from influenza. Since nursing students in both English studies consider vaccination unnecessary, it is essential to develop effective interventions in order to convince them otherwise.

Italy has the same influenza vaccination programs as France and the UK. Regarding the University of Palermo, although 62.53% of the nursing students who participated in a cross-sectional study were declared to be at high risk of contracting the flu, only 21.09% of them had been vaccinated for the flu season 2018/2019. A total of 188 out of 403 nursing students (46.65%) intended to be vaccinated against influenza during the following season (2019/2020) [28]. The low vaccine uptake of nursing students for the same flu season was also confirmed by another study in Milan (30.7%) [12]. However, during the 2018–2019 hospital campaign in Rome, the nursing and midwifery students actively participated in vaccination sessions, resulting in a notable vaccine uptake of 57.4% [13]. The above information underlines a significant disparity between how Italian nursing students perceive the risk of influenza and their actual utilization of the influenza vaccine. This inconsistency implies that factors other than risk perception, such as accessibility, convenience, and knowledge, play a role in their decision-making process regarding vaccination. Despite having vaccination policies comparable to France and the UK, nursing students in Italy exhibit low vaccination rates, indicating that country-specific elements may be influencing this matter. These elements could encompass cultural beliefs surrounding vaccination, characteristics of the healthcare system, educational campaigns aimed at healthcare professionals, and also increase the feeling of social responsibility. In order to effectively prevent the spread of communicable diseases and promote public health, it is crucial for every country to carefully consider these elements.

## 5. Limitations

Conducted as a cross-sectional survey, the present study gathered data that pertains to a specific moment in time—the completion of the survey questionnaire. This moment serves as a snapshot of the participant’s knowledge, attitudes, and practices related to influenza. It is important to note that while most scientific studies examining the vaccination coverage of nursing students about influenza are cross-sectional, they do not establish a clear temporal relationship between the determinant and the outcome. Furthermore, the sampling technique used in this study had limitations, as the convenience sample restricts the generalization of the findings to the entire population of nursing students. Additionally, the tool utilized in this research was a self-report questionnaire, which, despite being standardized, is subject to the limitations inherent in subjective evaluations. Therefore, further research conducted across multiple centers will provide additional insight into the aforementioned issues.

## 6. Conclusions

To conclude, this is the first large-scale, cross-sectional study of nursing students in Greece that investigated self-reported flu vaccination coverage, level of knowledge, attitudes, and practices against the disease. Participating students exhibited low vaccination rates. The findings suggest that their knowledge about vaccines significantly influenced their attitudes towards vaccination and subsequent uptake. Interestingly, protecting family members from influenza emerged as a primary motivation for those who did choose to vaccinate. The emphasis placed on social responsibility, family and patient protection, and well-being may be particularly effective with this population. Overall, data consistently demonstrate that nursing curricula are effective in developing informed HCWs who understand the importance of influenza prevention. Regular knowledge assessments among nursing students can identify knowledge gaps, allowing for targeted educational interventions. The implementation of continuing education programs, as well as the active involvement of students in the on-site vaccination of their colleagues, are necessary and essential interventions to increase the vaccination coverage of nursing students. Moreover, providing accessible vaccination opportunities, such as free clinics and on-campus immunization programs, must be expanded with the input of community nurses, who have the expertise to undertake this work. Ultimately, fostering a culture of vaccination among healthcare professionals as well as a well-informed and vaccinated population could help prevent influenza outbreaks and safeguard public health.

## Figures and Tables

**Table 1 diseases-12-00183-t001:** Socio-demographic characteristics of nursing students.

	N (%)
Gender	
Male	267 (21.2)
Female	993 (78.8)
Age (years), mean (SD)	21.9 (5.6)
Religion	
Christian	1118 (88.8)
Muslim	14 (1.1)
Jehovah’s Witness	4 (0.3)
Other	123 (9.8)
Nationality	
Greek	1172 (93.2)
Albanian	56 (4.5)
Romanian	7 (0.6)
Cypriot	14 (1.1)
Other	9 (0.7)
Family status	
Married	67 (5.3)
Single	1105 (87.8)
Divorced	5 (0.4)
Widowed	1 (0.1)
Living with partner	81 (6.4)
Working status	
Full time employed	194 (15.4)
Part-time employed	298 (23.7)
Unemployed	768 (61)
Permanent residence	
Big town	906 (71.9)
Small town	184 (14.6)
Rural area/village	88 (7)
Island	82 (6.5)
Year of study	
1st year	314 (24.9)
2nd year	274 (21.7)
3rd year	356 (28.3)
4th year	287 (22.8)
Upper year	29 (2.3)
Smoking	306 (24,3)
Alcohol consumption	530 (42,1)
Insurance capability	
No	63 (5)
Yes	907 (72)
I don’t know/I don’t answer	290 (23)
Private insurance	
No	709 (56.3)
Yes	228 (18.1)
I don’t know/I don’t answer	322 (25.6)
Do you trust the reliability of the information provided by the Ministry of Health about vaccinations?	
I don’t know/I don’t answer	16 (1.3)
I don’t trust at all	22 (1.7)
I have some reservations	425 (33.7)
I trust	646 (51.2)
I trust completely	152 (12.1)

**Table 2 diseases-12-00183-t002:** Nursing students’ knowledge of influenza.

Question	Answer	Ν	%
What are the common complications of the influenza virus?	Pneumonia	539	42.7
	Exacerbation of chronic bronchitis	370	29.3
	Otitis	219	17.4
	Sinusitis	288	22.8
	Exacerbation of diabetes mellitus	28	2.2
	*All of the above*	564	44.7
	I don’t know/I don’t answer	105	8.3
What are the usual symptoms of the influenza virus?	Fever	643	51
	Drowsiness/Depressive mood	446	35.4
	Refusal to receive food/fluids	284	22.5
	Difficulty in breathing	436	34.6
	Cyanosis	436	34.6
	*All of the above*	559	44.3
	I don’t know/I don’t answer	559	44.3
Which of the following groups of people do you consider to be at high risk of getting the flu?			
Patients with cancer	No	858	68.0
	*Yes*	403	32.0
Immunosuppressed	No	800	63.4
	*Yes*	461	36.6
Pregnant	No	946	75.0
	*Yes*	315	25.0
Brestfeeding	No	1133	89.8
	*Yes*	128	10.2
Postpartum women	No	1146	90.9
	*Yes*	115	9.1
Infants <6 months	*No*	878	69.6
	Yes	383	30.4
People with a body mass index (BMI) > 40 kg/m	No	1094	86.8
	*Yes*	167	13.2
Confined populations (e.g., prison inmates)	No	1131	89.7
	*Yes*	130	10.3
All of the above	*No*	565	44.8
	Yes	696	55.2
I don’t know/I don’t answer	*No*	1208	95.8
	Yes	53	4.2
Can the influenza virus be transmitted by droplets?	No	19	1.5
	*Yes*	1207	95.7
	Do not know	35	2.8
Can the influenza virus be transmitted through physical contact (e.g., shaking hands)?	No	401	31.8
	*Yes*	775	61.5
	Do not know	85	6.7
Can the influenza virus be transmitted through contact with a contaminated surface (e.g., a door handle)?	No	182	14.4
	*Yes*	977	77.5
	Do not know	102	8.1
Could influenza have serious and irreversible effects on the health of the general population?	No	193	15.3
	*Yes*	811	64.3
	Do not know	257	20.4
Can someone have the flu and be asymptomatic?	No	136	10.8
	*Yes*	909	72.1
	Do not know	216	17.1
What percentage of the general population gets the flu each year?	<1%	33	2.6
	*1–5%*	431	34.2
	>5%	796	63.2

Correct answers are in italics.

**Table 3 diseases-12-00183-t003:** Attitudes and practices on influenza.

Question	Answer	N	%
Would you advise a pregnant woman (regardless of gestational age) to get the flu shot?	No	178	14.1
	Yes	799	63.4
	Do not know	284	22.5
Would you get the flu shot if it were free?	No	162	12.8
	Yes	967	76.7
	Do not know	132	10.5
If the flu vaccine cost 10 euros, would you get it?	No	332	26.3
	Yes	779	61.8
	Do not know	150	11.9
If you were in a high-risk group, would you get the flu shot?	No	44	3.5
	Yes	1101	87.3
	Do not know	116	9.2
Do you avoid nursing patients with the flu?	No	1132	89.8
	Yes	72	5.7
	Do not know	57	4.5
Have you participated in an influenza-related education program in the past 12 months?	No	1135	90
	Yes	78	6.2
	Do not know	48	3.8
Do you use gloves in manipulations with an increased possibility of contact with the patient’s bodily fluids?	No	16	1.3
	Yes	1205	95.6
	Do not know	40	3.2
Do you use a surgical or other protective mask to enter a flu patient’s ward?	No	24	1.9
	Yes	1197	94.9
	Do not know	40	3.2
Do you consider it a problem to hospitalize patients with the flu?	No	1034	82
	Yes	176	14
	Do not know	51	4
If you answered “Yes” to the above question, please specify the reason.	Personal security	75	42.9
	Insufficient knowledge	7	4
	Inadequate protective measures	16	9.1
	High risk of infecting other people	77	44
Where do you think patients with the flu should be hospitalized?	Special Nursing Institutions	36	2.9
	Special Nursing Wards	401	31.8
	Isolation Wards	430	34.1
	Common Wards	228	18.1
	At home	165	13.1
What is your personal belief regarding your risk of contracting the flu?	No risk	27	2.1
	Low risk	202	16
	Moderate risk	704	55.9
	High risk	279	22.1
	I don’t know/I don’t answer	48	3.8

**Table 4 diseases-12-00183-t004:** Multiple logistic regression results of flu vaccination.

	OR (95% CI) +	*p*
Working Status		
Part-time employed vs. unemployed	0.65 (0.44–0.98)	0.042
Full-time employed vs. unemployed	0.44 (0.31–0.63)	<0.001
Do you trust the reliability of the information provided by the Ministry of Health about vaccinations? ^1^	1.58 (1.30–1.92)	<0.001
Have you participated in an influenza-related education program in the past 12 months? Yes vs. No	2.35 (1.44–3.83)	0.001

^1^ Answers could range from 1 (“not at all”) to 4 (“completely”); + odds ratio (95% confidence interval).

**Table 5 diseases-12-00183-t005:** Participants’ year of study and knowledge scores regarding influenza compared to their attitudes and practices against the disease.

	β +	SE ++	b ÷	*p*
**Year of study**				
2nd year	0.022	0.011	0.065	0.050
3rd year	0.039	0.011	0.126	<0.001
4th or greater year	0.032	0.012	0.101	0.003
**Knowledge score regarding influenza**	0.002	0.000	0.224	<0.001

+ dependence coefficient, ++ standard error, ÷ standardized coefficient.

## Data Availability

The data presented in this study are available on reasonable request from the corresponding author due to ethical requirements.

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
