# Peer review of "Influenza Vaccination of Nursing Students: A Cross-Sectional Study of Uptake, Knowledge, Attitudes, and Practices in Greece"

_diseases, 2024, doi:10.3390/diseases12080183_

Round 1
Reviewer 1 Report
Comments and Suggestions for Authors
A very interesting and fine study about what brings students (most of all nursing students) to get vaccinated against flu. It is noteworthy that the main factors determining acceptance of vaccination are (1) concern about protecting one's family, (2) adequate education about flu and vaccination. It is also important to note that mandatory vaccination did not bring the same results.
So, the conclusion for public health authorities is that they should invest in health education. (Concern about one's families health may not be universal; it may drop in more developed countries, where people may be more individualistic than in Greece. But education should be the main pont in getting populations, beginning with health workers, protected againts flu and many other diseases).
Lastly, if the authors deem it proper, I would advise them to enrich the Conclusions, since the material they have drawn may support more recommendations for health authorities. But the paper is sound, and can be published as it is.
Author Response
Dear Reviewer,
We would like to express our sincere gratitude for your thoughtful and constructive comments on our manuscript, "Influenza vaccination of nursing students: A cross-sectional study of uptake, knowledge, attitudes, and practices in Greece". Your insightful feedback has been invaluable in improving the quality of our work.
We have carefully considered each of your comments and have made significant revisions accordingly.
Comment 1: “Lastly, if the authors deem it proper, I would advise them to enrich the Conclusions, since the material they have drawn may support more recommendations for health authorities. But the paper is sound, and can be published as it is”.
Response 1
Thank you for pointing this out. We enriched the conclusions with more recommendations for health authorities as you advised us and we believe that these modifications have strengthened the overall clarity, rigor, and impact of our study. You can track the changes in the conclusion paragraph.
“To conclude, this is the first large-scale, cross-sectional study of nursing students in Greece that investigated self-reported flu vaccination coverage, level of knowledge, attitudes, and practices against the disease. Participating students exhibited low vaccination rates. The findings suggest that their knowledge about vaccines significantly influenced their attitudes towards vaccination and subsequent uptake. Interestingly, protecting family members from influenza emerged as a primary motivation for those who did choose to vaccinate. The emphasis placed on social rensponsibility, family and patient protection and well-being may be particularly effective with this population. Overall, data consistently demonstrate that nursing curricula are effective in developing informed HCWs who understand the importance of influenza prevention. Regular knowledge assessments among nursing students can identify knowledge gaps, allowing for targeted educational interventions. The implementation of continuing education programs as well as the active involvement of students in the on-site vaccination of their colleagues are necessary and essential interventions to increase the vaccination coverage of nursing students. Moreover, providing accessible vaccination opportunities, such as free clinics and on-campus immunization programs, must be expanded with the input of community nurses, who have the expertise to undertake this work. Ultimately, fostering a culture of vaccination among healthcare professionals as well as a well-informed and vaccinated population could help in preventing influenza outbreaks and safeguarding public health.”
We are confident that the revised manuscript is now of a higher standard and better reflects the valuable contributions of your review. Thank you again for your time and expertise.
Sincerely,
Our research team

Reviewer 2 Report
Comments and Suggestions for Authors
Review report
Title of the article: Influenza vaccination of nursing students: A cross-sectional 2 study of uptake, knowledge, attitudes, and practices in Greece
Abstract: The study aimed to investigate the vaccination coverage, knowledge, attitudes, and practices of nursing students against influenza in Greece. A cross-sectional study was conducted in the Attica region between September 2022 and July 2023, with the use of an anonymous reference questionnaire. The authors concluded that a higher level of knowledge about influenza was associated with more appropriate attitudes and practices toward the disease.
The abstract does show the main findings of the article. However, in view of the much wider spectre of questions applied in the questionnaire, I feel that it (the questionnaire) was largely unexplored in view of the little deepness with which the answers were interpreted.
Introduction: The introduction highlights a number of studies similar to theirs. However, is does it in a way that those are only merely presented, not commented or interpreted. No conclusions are presented with respect to the findings of those studies – in other words, the results are merely copied and not interpreted. This, in my view, in not adequate, since rather than presenting other authors’s results, the intro must highlight what was the outcome of those studies, or what do those mean. This resumes most of the introduction. Therefore I believe this must be reviewed and rewritten with a due interpretation of the findings of the authors referenced in the intro.
Methods.
The questionnaire: It covers a wide range of variables, what is impressive. It brings 68 questions which were commented and reviewed by qualified professionals.
Results
The results are the main part of the study. Unfortunately, the nicely prepared questionnaire was not fully explored. The results are mainly presented and not corrlated with the many variables that the questionnaire brings. No correlations or associations were drawn from many of the findings. For each of the variables, significance should be calculated, and these should be shown for each of those variables in the questionnaire, otherwise the meaning of asking such questions is lost.
A doubt: what is “lechoides”? I could not find the meaning of this…
Discussion
In the discussion, again many of the information gathered from the literature is merely resented, not discussed. Findings of others must be compared to those gathered in the study, so that conclusions can be drawn from it. Otherwise, the discussion becomes a mere presentation of data of others, the same problem found in the Intro section.
Limitations:
Here, the authors introduce HBV (?). It seems that this text was copied from some other article. Must be reviewed.
Conclusions:
Unfortunately, in view that the questionnaire was not fully evaluated in its potential, the conclusions gathered from the study are not far away from the obvious. This should also be reviewed after a full re-interpretation of the answers to the questionnaire.
Comments on the Quality of English LanguageThe text is quite understandable. A few minor corrections may be necessary.
Author Response
Dear Reviewer,
We would like to express our sincere gratitude for your thoughtful and constructive comments on our manuscript, "Influenza vaccination of nursing students: A cross-sectional study of uptake, knowledge, attitudes, and practices in Greece". Your insightful feedback has been invaluable in improving the quality of our work.
We have carefully considered each of your comments and have made significant revisions accordingly.
Comment 1: “Introduction: The introduction highlights a number of studies similar to theirs. However, is does it in a way that those are only merely presented, not commented or interpreted. No conclusions are presented with respect to the findings of those studies – in other words, the results are merely copied and not interpreted. This, in my view, in not adequate, since rather than presenting other authors’s results, the intro must highlight what was the outcome of those studies, or what do those mean. This resumes most of the introduction. Therefore I believe this must be reviewed and rewritten with a due interpretation of the findings of the authors referenced in the intro.”
Response 1:
Thank you for pointing this out. We improved the introduction section by adding a long paragraph in the third page, second last paragraph, lines 125-153. This paragraph interprets and comments on the findings of the authors referenced in the intro.
The above findings reveal a wide range of influenza vaccination rates among nursing students across different countries and flu seasons. In some cases, vaccination rates are notably low, such as in Poland (30.7% in 2019/2020) [5] and France (16.9% in 2011/2012) [15], while other countries like Sweden report higher rates (84.8% in 2009/2010) [7]. There are also noticeable differences in vaccination rates not only between countries but also within countries over time. For example, in Spain, the acceptance of the flu vaccine increased from 5.3% in 2014/2015 [8] to 83.4% in 2018/2019 [9], yet another study from the same year showed only 32.5% coverage among Madrid nursing students [10]. In Italy, rates varied significantly between cities and years, such as 21.09% in Palermo (2018/2019) [12] and 57.4% in Rome [13]. Despite the known risks associated with influenza, especially for HCWs, the data shows that many nursing students are not regularly vaccinated [8,12,20]. For instance, only 1.3% of nursing students in Poland were regularly vaccinated annually [5], and similar trends of low regular vaccination are observed elsewhere [8,9,15,20]. The reasons for low vaccination coverage are varied and include logistical barriers (e.g., difficulty finding someone to administer the vaccine in Spain [8]), perceptions of risk [9], and beliefs about the vaccine's importance [7-8]. In Greece, while there's no mandatory vaccination policy, the availability of free vaccines has not translated into high uptake, indicating potential cultural or educational barriers [20]. The findings reflect an ethical dimension, particularly concerning the students' intentions and preferences regarding patient safety. For instance, in Sweden, a significant portion of nursing students vaccinated themselves to protect patients, highlighting a professional ethic even among students [7]. The data also suggest that significant health events, such as the COVID-19 pandemic, may influence vaccination behavior. For example, the vaccination coverage among nursing students in Spain increased from 32.5% in 2019/2020 to 39% in 2021/2022 [11], possibly due to heightened awareness and concern about infectious diseases.
These findings suggest that while there is awareness of the importance of flu vaccination among nursing students, actual vaccination rates remain suboptimal in many regions. This discrepancy could be due to a combination of logistical, cultural, and educational factors.
Comment 2: “The questionnaire: It covers a wide range of variables, what is impressive. It brings 68 questions which were commented and reviewed by qualified professionals.”
Response 2:
Thank you for your consideration.
Comment 3: “The results are the main part of the study. Unfortunately, the nicely prepared questionnaire was not fully explored. The results are mainly presented and not correlated with the many variables that the questionnaire brings. No correlations or associations were drawn from many of the findings. For each of the variables, significance should be calculated, and these should be shown for each of those variables in the questionnaire, otherwise the meaning of asking such questions is lost.”
Response 3:
We have accordingly revised the results section of our paper as you advised us. More specifically, as many data as the survey could extract were added to the results section. All the necessary statistical analyses between dependent and independent variables were conducted by an experienced statistician, however, in this paper only the statistically significant findings are presented. We therefore added as many possible correlations were possible based to the statistical analysis performed by our research team.
In page 7, lines 276-281, data were added in order to justify the reasons that lead students to vaccination and the referred reasons of non-vaccination.
The most significant reason for influenza vaccination was personal protection (mean 1.2 ± 0.6), followed by the protection of the family environment (mean 1.9 ± 0.5), and lastly, the protection of patients (mean 2.7 ± 0.6). Conversely, the most significant reason for not getting vaccinated was not belonging to a high-risk group (mean 1.6 ± 1.0), followed by the belief that they would not get sick (mean 2.5 ± 0.9), adverse events (mean 2.8 ± 1.0), and lastly, vaccine efficacy (mean 3.1 ± 0.9).
A more detailed presentation of the data with an emphasis on the importance of the results was carried out in the following points of the results section: lines 315-318 on page 11.
Based on the correct responses regarding attitudes and practices towards influenza, percentage scores for recommended practices were calculated. In this sample, the minimum score was 0 and the maximum was 100. For recommended influenza practices, the mean was 77.6 ±18.6.
Additionally, four p-values were added on page 11, lines 310, 320, 321 and 322.
Also, higher trust in the reliability of the information provided by the Ministry of Health about vaccinations and having participated in an influenza-related education program in the past 12 months were statistically significantly associated with a greater probability of being vaccinated against influenza (p=0.001).
The score of appropriate attitudes and practices about influenza was significantly related to the protection of the nursing students' family environment (p=0.033), the mandatory vaccination in their workplace (p=0.001), and the outbreak of an epidemic (p=0.001).
Table 5, on page 12 was also added to more accurately present participants' year of study and knowledge scores regarding influenza compared to their attitudes and practices against the disease.
Table 5. Participants' year of study and knowledge scores regarding influenza compared to their attitudes and practices against the disease
|
|
β+ |
SE++ |
b÷ |
P |
|
Year of study |
||||
|
2nd year |
0,022 |
0,011 |
0,065 |
0,050 |
|
3rd year |
0,039 |
0,011 |
0,126 |
<0,001 |
|
4th or greater year |
0,032 |
0,012 |
0,101 |
0,003 |
|
Knowledge score regarding influenza |
0,002 |
0,000 |
0,224 |
<0,001 |
*+dependence coefficient, ++standard error, ÷standardized coefficient
Comment 4: “A doubt: what is “lechoides”? I could not find the meaning of this…”
Response 4:
The word "lechoides" was replaced with the tentative term "postpartum women", as you rightly pointed out. The changes can be tracked in page 8, line 284 and in Table 2 on page 9.
Comment 5: “In the discussion, again many of the information gathered from the literature is merely resented, not discussed. Findings of others must be compared to those gathered in the study, so that conclusions can be drawn from it. Otherwise, the discussion becomes a mere presentation of data of others, the same problem found in the Intro section.”
Response 5: We agree with your comment. In the discussion there was a thorough analysis and commentary of the findings of the study with other similar published research, in order to draw clearer conclusions as you pointed out. These changes can be tracked on page 12, lines 367-373 and 385-388, on page 13, lines 395-398, 401-403, 406-419, 430-435, on page 14 lines 447-454, 469-486, on page 15 lines 501-513, 519-524 and 531-544.
Comment 6: “Limitations: Here, the authors introduce HBV (?). It seems that this text was copied from some other article. Must be reviewed.”
Response 6:
In the limitations paragraph, lines 548 and 550, the term “HBV” was replaced with the term “influenza”, as you correctly identified an oversight.
Comment 7: “Conclusions:Unfortunately, in view that the questionnaire was not fully evaluated in its potential, the conclusions gathered from the study are not far away from the obvious. This should also be reviewed after a full re-interpretation of the answers to the questionnaire.”
Response 7:
Thank you for pointing this out. We enriched the conclusions with more recommendations and measures for health authorities as you advised us and we believe that these modifications have strengthened the overall clarity, rigor, and impact of our study. You can track the changes in the conclusion paragraph.
To conclude, this is the first large-scale, cross-sectional study of nursing students in Greece that investigated self-reported flu vaccination coverage, level of knowledge, attitudes, and practices against the disease. Participating students exhibited low vaccination rates. The findings suggest that their knowledge about vaccines significantly influenced their attitudes towards vaccination and subsequent uptake. Interestingly, protecting family members from influenza emerged as a primary motivation for those who did choose to vaccinate. The emphasis placed on social rensponsibility, family and patient protection and well-being may be particularly effective with this population. Overall, data consistently demonstrate that nursing curricula are effective in developing informed HCWs who understand the importance of influenza prevention. Regular knowledge assessments among nursing students can identify knowledge gaps, allowing for targeted educational interventions. The implementation of continuing education programs as well as the active involvement of students in the on-site vaccination of their colleagues are necessary and essential interventions to increase the vaccination coverage of nursing students. Moreover, providing accessible vaccination opportunities, such as free clinics and on-campus immunization programs, must be expanded with the input of community nurses, who have the expertise to undertake this work. Ultimately, fostering a culture of vaccination among healthcare professionals as well as a well-informed and vaccinated population could help in preventing influenza outbreaks and safeguarding public health.
Comment 8: “The text is quite understandable. A few minor corrections may be necessary.”
Response 8:
We have read the entire text very carefully and tried to correct any errors of expression and incorrect use of the English language.
We are confident that the revised manuscript is now of a higher standard and better reflects the valuable contributions of your review. Thank you again for your time and expertise.
Sincerely,
Our research team

Reviewer 3 Report
Comments and Suggestions for Authors
The investigation method of this research is scientific, the investigation results are clear, and the improvement methods and measures should be put forward to make the investigation play a greater role. For example: to inform medical students (doctors or nurses) about the effects of influenza and the effects of vaccines (lectures, publicity papers), so that they understand that annual immunization with influenza vaccines is to protect themselves, their families and patients, so as to affect more people (family, friends, neighbors and patients); The second is to take some necessary convenience measures for the specific matter of immunization against influenza vaccine, such as: conducting free clinics or voluntary activities when holding large-scale activities - immunizing the people with vaccines, and providing collective immunization against influenza for students in large, middle and primary schools (especially medical schools) every year.
Author Response
Dear Reviewer,
We would like to express our sincere gratitude for your thoughtful and constructive comments on our manuscript, "Influenza vaccination of nursing students: A cross-sectional study of uptake, knowledge, attitudes, and practices in Greece". Your insightful feedback has been invaluable in improving the quality of our work.
We have carefully considered each of your comments and have made significant revisions accordingly.
Comment 1: “Does the introduction provide sufficient background and include all relevant references?”
Response 1:
We improved the introduction section by adding a long paragraph in the third page, second last paragraph, lines 125-153. This paragraph interprets and comments on the findings of the authors referenced in the intro.
Comment 2: “the improvement methods and measures should be put forward to make the investigation play a greater role”.
Response 2:
Thank you for pointing this out. We enriched the conclusions with more recommendations and measures for health authorities as you advised us and we believe that these modifications have strengthened the overall clarity, rigor, and impact of our study. You can track the changes in the conclusion paragraph.
“To conclude, this is the first large-scale, cross-sectional study of nursing students in Greece that investigated self-reported flu vaccination coverage, level of knowledge, attitudes, and practices against the disease. Participating students exhibited low vaccination rates. The findings suggest that their knowledge about vaccines significantly influenced their attitudes towards vaccination and subsequent uptake. Interestingly, protecting family members from influenza emerged as a primary motivation for those who did choose to vaccinate. The emphasis placed on social rensponsibility, family and patient protection and well-being may be particularly effective with this population. Overall, data consistently demonstrate that nursing curricula are effective in developing informed HCWs who understand the importance of influenza prevention. Regular knowledge assessments among nursing students can identify knowledge gaps, allowing for targeted educational interventions. The implementation of continuing education programs as well as the active involvement of students in the on-site vaccination of their colleagues are necessary and essential interventions to increase the vaccination coverage of nursing students. Moreover, providing accessible vaccination opportunities, such as free clinics and on-campus immunization programs, must be expanded with the input of community nurses, who have the expertise to undertake this work. Ultimately, fostering a culture of vaccination among healthcare professionals as well as a well-informed and vaccinated population could help in preventing influenza outbreaks and safeguarding public health.”
We are confident that the revised manuscript is now of a higher standard and better reflects the valuable contributions of your review. Thank you again for your time and expertise.
Sincerely,
Our research team

Round 2
Reviewer 2 Report
Comments and Suggestions for Authors
I believe the paper may be considered for publication.